# TiSiCN as Coatings Resistant to Corrosion and Neutron Activation

**DOI:** 10.3390/ma16051835

**Published:** 2023-02-23

**Authors:** Matlab N. Mirzayev, Anca C. Parau, Lyubomir Slavov, Mihaela Dinu, Dimitar Neov, Zdravka Slavkova, Evgeni P. Popov, Maria Belova, Kanan Hasanov, Fuad A. Aliyev, Alina Vladescu (Dragomir)

**Affiliations:** 1Joint Institute for Nuclear Research, 141980 Dubna, Russia; 2Institute of Radiation Problems, Azerbaijan National Academy of Sciences, AZ1143 Baku, Azerbaijan; 3Scientific-Research Institute Geotecnological Problems of Oil, Gas and Chemistry, Azerbaijan State Oil and Industry University, AZ1010 Baku, Azerbaijan; 4Innovation & Research Center, Western Caspian University, AZ1001 Baku, Azerbaijan; 5National Institute of Research and Development for Optoelectronics INOE 2000, 409 Atomistilor St., 77125 Magurele, Romania; 6Institute of Electronics, Bulgarian Academy of Sciences, 1784 Sofia, Bulgaria; 7Institute of Solid-State Physics, Bulgarian Academy of Sciences, 1784 Sofia, Bulgaria; 8Institute for Nuclear Research and Nuclear Energy, Bulgarian Academy of Sciences, 1784 Sofia, Bulgaria; 9Institute of Geology and Geophysics of Azerbaijan, National Academy of Sciences, AZ1143 Baku, Azerbaijan; 10Physical Materials Science and Composite Materials Centre, Research School of Chemistry & Applied Biomedical Sciences, National Research Tomsk Polytechnic University, 30 Lenina Avenue, 634050 Tomsk, Russia

**Keywords:** carbonitrides, corrosion resistance, surface morphology, neutron activation analysis, crystal structure, potentiodynamic polarization

## Abstract

The aim of the present paper was to evaluate the effect of neutron activation on TiSiCN carbonitrides coatings prepared at different C/N ratios (0.4 for under stoichiometric and 1.6 for over stoichiometric). The coatings were prepared by cathodic arc deposition using one cathode constructed of Ti88 at.%-Si12 at.% (99.99% purity). The coatings were comparatively examined for elemental and phase composition, morphology, and anticorrosive properties in 3.5% NaCl solution. All the coatings exhibited f.c.c. solid solution structures and had a (111) preferred orientation. Under stoichiometric structure, they proved to be resistant to corrosive attack in 3.5% NaCl and of these coatings the TiSiCN was found to have the best corrosion resistance. From all tested coatings, TiSiCN have proven to be the most suitable candidates for operation under severe conditions that are present in nuclear applications (high temperature, corrosion, etc.).

## 1. Introduction

In the last few years, ceramics gained more popularity in nuclear fusion applications due to their high insulation properties, heat, and radiation resistance when compared to other commonly used isolators (i.e., plastics). Ceramic materials can be used for the elements of plasma facing (i.e., in diverters and as insulation coating of coils) or for the construction components of tokamaks. In the first case, the materials are permanently exposed to different types of ionizing radiation generated inside the fusion chambers and they should possess certain properties; on the other hand, in the second case, the irradiation will be applied just in case of emergency and the properties should be different than those necessary for constant exposure.

In the present paper, our attention is focused on materials subjected to continuous irradiation. This dangerous phenomenon is associated with the ion component of the internal tokamak plasma due to the deceleration length of ions inside the plasma-facing solid materials, which is short in comparison with that of the other disturbing radiations (i.e., X-ray photons, electrons, and neutrons). In the tokamak plasmas, two magnitudes of ion energies are reachable: plasma temperature (about 10 keV) and the beams of fast ions produced during neutral-beam plasma heating (about 100 keV). The diffusion through an insulating magnetic field of plasma inside a tokamak chamber is transformed into the plasma temperature of about <1 keV, while the energy of fast ions bombarding the plasma-facing elements becomes very wide; thus, based on what we presented above, we believe that the testing of the radiation effect on various types of ceramics is essential.

By comparing the coatings prepared by physical vapor deposition (PVD) methods, it can be seen that carbonitrides are widely used for applications where mechanical, tribological, and oxidation resistance are needed. Carbonitride coatings consist of “perfectly” mixed C and N in a f.c.c structure and combine the best characteristics of both parent components (carbide and nitride). For example, TiCN coatings provide superior mechanical resistance and thermal stability as compared to independent TiC and TiN coatings. That is due to the combination of the high ductility and the high melting point of TiC, and to the superior adhesion strength and the low internal stress of TiN [1]. Moreover, there are several configurations accessible for the carbonitrides, such as monolayer, multilayer, or graded structure, respectively, offering the possibility to alter the structure and composition in a precise way leading to adjustable properties [2]. Among all carbonitrides, those of ternary coatings containing one transition metal, such as TiCN, ZrCN, NbCN, or CrCN, are most widely used for industrial applications [3,4,5,6]. Even though TiCN and CrCN have already become regular commercial products, extensive research is still needed. In the last years, a new generation of carbonitrides was proposed by the addition of various metallic and/or non-metallic elements into the basic ternary matrix. For example, Constantin et al. reported that inserting small amounts of Zr, Nb, or Si into TiZrCN, TiNbCN, and TiSiCN systems leads to stress decrease and adhesion increase; also, the corrosion processes of high-speed steel in NaCl aggressive environment were significantly improved [7]. Up to this date, some complex Ti carbonitrides coatings prepared by PVD were studied (Cr and/or Si were used as alloying elements in structures such as TiCrCN [8], TiSiCN [9,10,11], TiCrNbCN [12], TiAlSiCN [13,14,15], TiCrSiCN [15,16,17], and TiNbCN [18,19,20]), revealing their superior properties for different applications.

The TiSiCN coatings were selected due to their known properties, such as high hardness, low friction, and wear performance [21]. Up to this date, TiSiCN coatings were prepared by magnetron sputtering [22] or CVD methods [9,11,23]. It was reported that their structure and mechanical characteristics were influenced by the Si and C contents in the composition [24]. For the TiSiCN coatings prepared by magnetron sputtering, many researchers reported that their strength is due to “nanocrystallite (nc) and amorphous (a) phases” interface [9,22,23,25,26,27,28,29]. This model was proposed by Veprek in 1995 for the “nc-TiN/a-Si_3_N_4_” coatings to describe the strengthening mechanism of the TiSiN nanocomposites [30]. Nevertheless, the topic related to the interface of “nc-nanocrystallite/a-amorphous” system is still under heated discussion due to the experimental confirmation deficiency for the interfaces state—whether it is amorphous or crystalline. Regarding the TiSiCN coatings prepared by CVD, it was reported that the performance, such as oxidation resistance, improved by increasing the Si content [31]; however, very limited information related to the corrosion resistance of TiSiCN coatings and the preparation by cathodic arc technique could be found in the literature [32,33,34].

To our knowledge, none of the mentioned carbonitrides have been used for nuclear fusion applications and there is no available information about the irradiation behavior of these materials. ZrC, TiC, AlN, ZrN, SiN, and TiN have been considered as inert coatings suitable for fusion reactors [35], but the existing information is rather about the effect of irradiation on these materials.

In the present paper, the corrosion behavior represents the subject of analysis, because of the high corrosion attack appearing in nuclear fusion and, therefore, coatings with high corrosion barriers are needed. For example, Konys et al. demonstrated that severe corrosion dissolved steel components by forming precipitates which blocked the system [36]. Moreover, the corrosion leads to a loss of mechanical integrity of materials. Thus, knowledge related to corrosion behavior of materials used in nuclear fusion is very important. For example, Cr based alloys are commonly used for molten salt reactors and information about the effect of Cr amount variation in the alloy and temperature on the corrosion of materials in the chloride salts is not clear. Further systematic studies should be well conducted for the understanding of the chloride salt corrosion mechanism. The challenges about the electrochemical processes in the molten salt reactor are considered to be relatively innovative and few papers on this subject are published; thus, the corrosion of materials is a more challenging topic in the case of molten salt nuclear systems compared to traditional water reactors due to the formation of a passive oxide layer on top of the surface, which is thermodynamically unfavorable in molten salts, leading to a restriction of materials is this area.

The TiSiCN coatings deposited on Ti6Al4V substrate were not so much discussed in the literature, despite their excellent wear and corrosion resistance suitable for many industrial applications, such as navigation, aerospace, engine turbine, and ocean exploration. Wang et al. reported that the C/N ratio in TiSiCN coatings obtained by arc ion plating has a great influence on the tribocorrosion resistance in artificial seawater (standard ASTM D1141-98) [24].

The aim of the present paper is to study the effect of neutron activation on TiSiCN carbonitride coatings prepared by the cathodic arc evaporation method in a reactive gas mixture of C_2_H_2_ and N_2_. The coatings were obtained at different C/N ratio (0.4 and 1.6) to study the effect of C/N ratio variation on the neutron activation. The obtained coatings are investigated in terms of elemental and phase composition, crystalline structure, morphology, and corrosion resistance in aggressive saline solution (3.5% NaCl), in order to be considered suitable candidates for operation under severe conditions present in nuclear applications.

## 2. Materials and Methods

### 2.1. Deposition of the Coatings

The cathodic arc deposition system was used for the preparation of TiSiCN on Ti6Al4V alloy. The unit was equipped with one cathode constructed of Ti 88 at.%-Si 12 at.% (99.99% purity). The base pressure prior to coating was of 6 × 10^−4^ Pa; the samples were biased at −1000 V for 10 min in an Ar atmosphere at a pressure of 0.2 Pa. For each deposition the Si content was carefully adjusted up to 6 at. % such as the final (C + N)/(metal + Si) ratio to range between 0.6 and 0.8. Depending on the C/N ratio, under stoichiometric coating (0.4) is labeled TiSiCN-1 and over stoichiometric (1.6) as TiSiCN-2. The gas mass flow rates for TiSiCN-1 coating were kept at 25 sccm for CH_4_ and at 65 sccm for N_2_; for TiSiCN-2 the gas flow rates were inversed. Arc current was at 110 A and the substrate bias was −100 V. All parameters were kept constant for maximum 40 min, leading to a deposition temperature of 320 °C and thickness of the coatings of ~3 μm. More preparation details could be found in [7,34].

### 2.2. Corrosion Testing and Characterization of the Coatings

The corrosion resistance has been evaluated by the polarization technique in 3.5% NaCl (purchased from Sigma-Aldrich, Hamburg, Germany), using a potentiostat/galvanostat (VersaStat, Princeton Applied Research, Oak Ridge, TN, USA). A typical three-electrode cell was used: sample as working electrode (WE), platinum foil used as counter electrode (CE), and saturated Ag/AgCl-electrode as reference electrode (RE). The open circuit potential (OCP) was monitored until equilibrium occurred. The potentiodynamic curves were plotted from −1 V to +2 V vs. OCP. Corrosion potential (*E*_corr_) and corrosion current density (*i_corr_*) parameters were extracted from the Tafel extrapolations in the range of ±50 mV. Polarization resistance (*R*_p_) was calculated according to the procedure described in ASTM G59-97 standard (reapproved 2014). The corrosion rate (CR) has been calculated by the following formula, assuming that uniform corrosion has taken place over the whole immersed surface:(1)$$\mathrm{CR}\;\frac{\mathrm{mm}}{\mathrm{year}}=\mathrm{weight}\;\mathrm{loss},\;g\times \frac{{\mathrm{cm}}^{3}}{\mathrm{material}\;\mathrm{density},\;g}\times \frac{1}{\mathrm{exposed}\;\mathrm{area},{\;\mathrm{cm}}^{2\;}}\times \frac{10\;\mathrm{mm}}{1\;\mathrm{cm}}\times \frac{1}{\mathrm{exposed}\;\mathrm{time},\;h}\times \frac{8760\;h}{1\;\mathrm{year}}$$

The anti-corrosive properties of the coatings were estimated by calculating the protective efficiency (*P_e_*) [37]:(2)$$P_{e}=\left(1-\frac{i_{corr,\;coating}}{i_{corr,\;substrate}}\right)\times 100$$
where *i_corr,coating_* and *i_corr,substrate_* are the corrosion current densities of coating and substrate, respectively.

The total porosity (*P*) of a coating was assessed using Elsener’s empirical equation [37]:(3)$$P=\left(\frac{R_{p,\mathrm{substrate}}}{R_{p,\mathrm{coating}}}\right)\times 10^{\frac{-\left|\Delta E_{\mathrm{corr}}\right|}{\beta_{a}}}$$
where *R*_p,substrate_ and *R*_p,coating_ are the polarization resistances of the substrate and coating, respectively, Δ*E*_corr_ represents the difference between the corrosion potentials of the coatings and substrate, and *β*_a_ is the anodic slope of the Tafel extrapolation of the substrate.

The EIS measurements were performed over 0.5 ÷ 10^3^ Hz frequency range by applying a sinusoidal signal of 10 mV RMS vs. EOC. Data were recorded by VersaStudio software and the fitting procedure was performed using ZView software.

The morphology of the surface before and after corrosion tests was investigated using a scanning electron microscope (Hitachi 3030PLUS, Tokyo, Japan). The surface roughness before and after corrosion tests was evaluated using a surface profilometer (Dektak 150, Bruker, Tucson, Arizona) with a stylus diameter of 2.5 µm over a length of 10 mm on two replicates in 10 randomly selected different areas in the centre of each sample. *R*_a_ (arithmetic average) and S_k_ (profile symmetry relative to the mean line) parameters were used to estimate the roughness of the investigated surfaces.

The elemental composition of both TiSiCN coatings was investigated using an equipped energy dispersive X-ray detector (EDS, Bruker) and a scanning electron microscope (Hitachi 3030PLUS). The crystalline structure and the phase composition were obtained by means of X-ray diffraction (XRD) using a Rigaku SmartLab diffractometer, with Cu Kα radiation (1.5405 nm) set up in θ/2θ geometry range of 30–80° with a step of 0.02°/min. Additionally, grazing incidence XRD (GRXRD) was performed by fixing the incident angle at 3°. The phase identification was aspired by Rietveld profile fitting using the FullProf program and the crystallite sizes were determined from the XRD peak widths using the Scherrer formula.

### 2.3. Fast Neutrons Irradiation and the Neutron Activation Analysis (NAA)

The neutron irradiation of the prepared samples was performed at the IBR-2M pulsed reactor at the Frank Laboratory of Neutron Physics (JINR, Dubna, Russia). The conditions during irradiation were the nominal power of 1450 kW (average power of the reactor during the period cycle) and the cycle time of 288 h. The neutron activation analysis was used for the determination of fast neutrons flux density (by nickel) and fluence on the specimens. The investigated isotope (line in the spectrum) was ^58^Co (810.7 keV) and the accepted effective section was 0.092 barn. Laboratory gamma-spectrometer Canberra GC10021 and Lynx multichannel analyzer were used to conduct NAA with high accuracy.

## 3. Results and Discussions

### 3.1. Elemental Composition Surface Morphology (EDS)

In Table 1, results of the EDS elemental composition analysis of the deposited coatings are shown. Synthesized under different conditions (C/N ranged from 0.4 to 1.6 stoichiometric and C + N(/(Metal + Si) ranged 0.6 to 0.8), the base elements (Ti, Si, C, and N) are predominant in research samples.

### 3.2. X-ray Diffraction Analysis of TiSiCN

For confidence, in order to distinguish reflections belonging to the substrate and the coatings, the substrate without coating was measured alone (Figure 1a). The main substrate peaks were indexed as (100), (002), (101), (102), (110), (103), (112), and (201). Titanium alloy reflection (200) was also present, but with a lower intensity. Most of the reflections were indexed in space group: P63/mmc (α-Ti, hexagonal close packed). The lattice parameters proved to be a = b = 2.93706 Å and c = 4.69061 Å. According to R. Pederson [38], some residual amount of β phase (SG Im$\bar{3}$m) should remain and the peak at 2θ = 39.67° could be indexed as (110) of β-Ti. One can notice that the texture of the substrate is extreme, which is not surprising since Ti alloys tend to exhibit a very strong texture after the hot-rolling manufacturing process. Usually, it is described as {11-20}(0001). In our case, we find (10-10) normal component to be the strongest. The texture was considered as fiber described by FullProf incorporated exponential function P_h_ = G_2_ + (1-G_2_)exp(G_2_α_2_) with texture parameter G_2_ > 7 (from FullProf). The texture was very strong with main component (10-10), which diverges from (0001) mentioned in the literature. This, however, is perfectly possible, since we are unaware of the direction of cutting from the ingot.

The scattering density of the substrate, Ti6Al4V, is considerably higher than that of the coating; therefore, the superposition of substrate reflections was expected. Upon inspection of data from the TiSiC-1 sample (Figure 1b), it can be seen that the spectrum displayed complexity with superposition of individual phases of different features. The spectrum was characterized by much sharper peaks presumably belonging to Ti6Al4V substrate. Clearly, several substrate peaks emerged, at 2θ = 42.00°, 60.4°, and 72°; also, a broadened peak (2θ = 36°) superimposing (100) Ti6Al4V was observed. These positions coincide roughly with the peaks of the TiN structure found in the literature. The space group was Fm$\bar{3}$m (cubic) and the lattice parameter a = 4.31263 Å; however, inside the coating, the lattice appears distorted, with lattice parameter slightly above the literature value of 4.28 Å. One could assume that since TiC possesses the same space group, but higher a TiC phase is present in the coating, but the value for TiC falls out of the range of possible a. Therefore, more probable is the situation where the composition of the coating is not fixed, but C/N ratio has some continuous distribution and, hence, results in continuous dispersion of the cell dimensions (spanning). This novel result is reported for the first time. All these peaks are much broadened, and an attempt was made to estimate the size of coherent scattering blocks using Scherrer equation from (111) peak width. It yields upper limit value for d = 10 nm, which is the evidence for highly dispersed nanostructured coating. The specimen featured very sharp axial texture with overwhelming component (111) in diffraction spectrum; therefore, direction [111] is normal to the surface, with FullProf texture exponential parameter equals 9. The strength of the texture is a very strong texture component.

In the case of C/N ratio 1.6 (TiSiCN-2), it is expected that TiC will prevail in the coating. TiC crystallizes in the same group Fm$\bar{3}$m. XRD of the coating with predominant carbon content was shown in Figure 1c. An effect was observed, which is the slight shift of the coatings peaks towards lowest angles that indicated a content of TiC with the same crystal symmetry, but smaller unit cell parameters than TiN and correspondingly higher diffraction angles. The second difference is a drastic intensity decrease in (111) the reflection in comparison with TiN case and, hence, the decrease in the sharp (111) texture to a negligible one (FullProf texture parameter 0.12470). Lattice parameter was found a = 4.32 Å.

The comparison of XRD spectra before and after corrosion procedure in saline solution did not show any significant changes; however, for TiSiCN-2, we observe considerable decrease in intensities of the diffraction peaks. It could be that some, although little, X-ray intensity is taken away by the oxide layer formed on top of the surface. XRD spectra of grazing geometry showed an intriguing phenomenon for the investigated coatings (see Figure 2). In this case, the diffraction power decrease is much stronger than in the Bragg–Brentano geometry case and this time it mainly concerns diffraction from the coatings. The peak erosion is stronger in TiSiCN-2 than in TiSiCN-1, meaning under stoichiometric coating, which is more corrosion-resistant than the over stoichiometric, i.e., nitrogen has a positive influence on the chemical protection of the coating. All peaks are eroded after corrosion, but the effect manifests itself better on the peak at 2θ = 39.6°. The same is observed on the TiSiCN-2. The same peak exhibits the strongest decrease. We suspect that this reflection, although on the place of β-Ti (110), does not belong to it but to TiO_2_-rutile phase (200) synthesized during the high-temperature coating deposition. Rutile, although not soluble in saline solution, apparently has worse adhesion with the rest of the coating layer and is washed away during the corrosion treatment.

### 3.3. Surface Morphology

The surface morphology images (SEM), alongside the elemental mapping profiles of the TiSiCN coatings on Ti6Al4V alloy, before and after corrosion testing in 3.5% NaCl are presented in Figure 3.

Many micro-droplets with different diameters could be seen on the surfaces of the deposited pre-corrosion coatings (Figure 3a,b). The appearance of such micro-sized morphology features is typical not only for the cathodic arc technique but is observed as well for the same coatings that are fabricated using PEMS, CVD, or arc ion plating methods [24,39,40,41,42]. They are unavoidable and generated during the deposition process and tend to be more pronounced in thicker coatings [40]. From the figures, it could be seen that the micro-droplets differ in size and surface density. On the surface of TiSiCN-2, more and larger droplets are observed, with some reaching up to 20 μm in size. That could be in connection with the difference in the gas mass flow rates (CH_4_ and N_2_) in both samples during deposition, since intensity of defects and growth-induced stress are strongly dependent on the process parameters [43,44].

An assumption could be made regarding the formation process of such microdroplets in the samples under study here. In many papers, it is reported that TiSiCN coating consists predominantly of TiC_x_N_y_ nanosized crystals embedded in SiC_x_N_y_)/a-C amorphous matrix [11,40,41]. It is possible that the lesser amount of nitrogen in the TiSiCN-2 system compared to TiSiCN-1 hinders the amorphous matrix formation, which is reported to play a major role in the nanocrystal grain growth, since it is an interface phase [9,45]. That could lead to more nucleation sites, larger particles to evolve (microdroplets), and also a greater likelihood of occurrence of inter-particle interactions; so, a nitrogen deficiency could lead to more vacant Ti sites available for oxidation and formation of a TiO2 layer on top, as illustrated in [46]. The EDS data for TiSiCN-2 (Table 1) somewhat reinforce such an assumption—lower nitrogen content coincides with increased percentage values for both Ti and O.

It should be noted that similar large surface formations have been shown to occur also on coatings without carbon (TiSiN) [46,47]. Studies on such formations reveal that they consist predominantly of Ti, N, and O [46,47]; thus, it seems that the different carbon content in the investigated TiSiCN samples should not play a role in the difference in both number and density of the microdroplets. On the other hand, it is reported that higher carbon content (as in TiSiCN-2) enhances the amorphous phase amount and favors reducing the size of crystallites [9,44]. On the carbon elemental profile map, some areas with increased concentration could be seen on both coatings (Figure 3a,b). In the titanium and nitrogen mapping profiles, the same areas lack in color, thus unveiling the absence of both elements. Since there is no titanium in these areas, the presence of carbide structures should be excluded and only the accumulation of amorphous carbon should be regarded in these locations. Again, it seems that the difference in carbon content in both coatings does not play a role in the formation of such carbon piles, since they are observed in both TiSiCN coatings.

Despite the presence of microdroplets, both coatings were uniformly deposited, without any major defects, such as pinholes, voids, or cracks. The different C/N ratio in the coatings is easily recognizable from the difference in density on C and N elemental profiles but does not have a major effect on the titanium homogeneity in the coatings. The smoother surface morphology in TiSiCN-1 (SEM images in Figure 3), could be associated with the bigger quantity of nitrogen, since it is essential for the formation of both TiN and Si_3_N_4_ amorphous matrix components.

Despite the aggressive corrosion medium, there was no change in the surface morphology. A decrease in the number of micro and macro particles of different sizes was observed only on the surface of the sample. The surface morphology of TiSiCN coatings can be due to carbon content, as previously reported [9]. Usually, these defects constitute preferential diffusion trails of aggressive species and can result in an accelerated coating failure [34]. In addition, Li investigated the surface morphology of TiSiCN coatings at the C/Si 4:0, 3:1, 2:2, 1:3, and 0:4 ratios and concluded the ideal surface morphology is formed in the ratio C:Si = 2:2 when no surface degradation occurs [21].

### 3.4. Surface Roughness

Table 2 contains the roughness parameters values before and after corrosion testing performed in 3.5% NaCl. R_a_ is the average arithmetic deviation from the mean line linked with the peaks profile height/depth and valleys present on the surface [48]. Both coated surfaces exhibited much higher R_a_ values than the uncoated substrate before and after corrosion investigation, indicating rougher surfaces. Such results were expected because of the observed morphology patterns in SEM images, including microdroplets and pores visible on the surface of both coatings. Again, in good agreement with SEM observations, TiSiCN-2 coating exhibits rougher surfaces than TiSiCN-1 coating before corrosion. It seems that the lower C/N ratio leads to smoother surface of the coating. Although the uncoated substrate showed its smoothest surface before corrosion (*R*_a_~37 nm), after the corrosion attack R_a_ decreased to 67.9% of its original value, indicating a severe corrosive process. On the other hand, the deviation in the R_a_ values for both coated surfaces of 2.4% and 0.2% for TiSiCN-1 and TiSiCN-2, respectively, indicate that the coatings were almost intact after corrosion testing, signifying a less pronounced corrosion of these surfaces. The difference in the R_a_ values of the two coatings is statistically insignificant; thus, it could be concluded that both are not prone to significant changes upon corrosion attack. It could also be noted that the TiSiCN-2 rougher surface seems to be less susceptible to corrosion, compared to the smoother surface of the TiSiCN-1 sample.

Considering the skewness value, it is recognized that a positive value indicates better corrosion resistance [49]. By comparing the S_k_ values before corrosion, one may see that the uncoated substrate has negative value, unlike the two film specimens, indicating that the surface has more valleys than peaks, probably due to the polishing process. After corrosion testing, the S_k_ value is more pronounced, displaying the depression depth increase, meaning that the corrosive solution dug deep in the initial cavities found on the Ti6Al4V surface. The corrosion experiment has a different effect on the two coatings. While for the TiSiCN-1 the S_k_ value lowers, meaning some kind of smoothing, with the filling of available pits and the peaks becoming less pronounced, in the case of the TiSiCN-2 coating an opposite process unfolds (Table 2). This suggests a possible difference in the chemical composition of the constituents of the surfaces of the two coatings. The S_k_ values were close before and after corrosion of TiSiCN-1; thus, it could be concluded that the corrosive solution did not affect the coating surface to a big extent. On the contrary, in the case of TiSiCN-2, a significant increase in S_k_ value after corrosion could be observed, meaning that a severe corrosive process has taken place on the surface. It is possible that a lower C/N ratio in TiSiCN-1 to make the surface more prone to passivation process and quick development of an oxide layer, consequently increasing its corrosion resistance. Films with higher C/N ratio are more abundant in microdroplets and they are larger. It results in a rougher surface, which leads to a larger interfacial area with the corrosive environment and, therefore, an enhanced rate of the corrosion process is expected [49].

### 3.5. Corrosion Behaviour

The EIS data obtained for TiSiCN-coated TiAlV alloy are presented in Figure 4 as both Nyquist and Bode amplitude plots. As observed in Nyquist impedance plots, slightly higher semicircles were obtained for TiSiCN-1 coating, indicating a higher charge transfer resistance. The main frequency semicircle is modeled as a combination of parallel resistance and capacitance elements in series with electrolyte resistance. The electrical equivalent circuit (EEC) used for fitting the impedance data is also presented as an insert in Figure 4a. Two interfaces were considered: the porous outer oxide layer/coating-electrolyte interface (characterized by CPE_coat_ and *R*_pore_ parameters) and the inner layer formed at the electrolyte/dense film interface (characterized by CPEdl and Rct parameters). The same EEC was used also for the Ti6Al4V substrate, the model being an indicated circuit used for coated surfaces. According to Pan et al., a bilayer structure of oxide film is formed on the surface of titanium during the immersion in a saline environment: TiO_2_ dense inner layer and porous Ti6Al4V alloy outer layer [50]. According to literature reports, the thickness and characteristics of the oxide film depends on the testing solution [50], but this is mainly formed of TiO_2_, Ti_3_O_5_, Ti_2_O_3_, and TiO oxides [51,52,53]. I. Milošev et al. used the same model to fit the impedance data taking into consideration the formation of the mentioned oxides and suboxides formed on the Ti-based alloys immersed in different media, as indicated by XPS analysis [51,54].

Table 3 presents the fitted parameters of impedance data obtained, along with *χ*^2^ parameter as an indication of the goodness of fit. For Ti6Al4V alloy, notice the high value of the *Q*_dl_ (resistance associated to the current flow through the pores), which shows that the formed oxide layer allows the electrolyte to ingress through the pores. Indeed, the pore resistance indicated the lowest value as compared with the coated surfaces (*R*_pore_ = 4 kΩ cm^2^). Instead, the highest value was showed by TiSiCN-1 coating (*R*_pore_ = 3626 kΩ cm^2^), indicating a more compact structure with beneficial effects on electrochemical behavior when immersed in 3.5% NaCl solution. This coating also showed the highest α_coat_ parameter (0.77), followed by TiSiCN-2 (0.63), which are values related to non-uniform current distribution along the surface [55]. At the interface with the substrate, there is an almost defect-free surface, since α_dl_ showed high values near 1; thus, the constant phase element (CPE) used in this case could be seen as an ideal capacitor. Instead, at electrolyte-coating interface α_coat_ showed low values and a CPE is needed to take into consideration possible deviations from the ideal dielectric behavior [56]. As pointed out in literature, there are multiple factors associated with these deviations, which include surface disorder or inhomogeneity, geometric irregularities, working electrode porosity, or ether surface roughness [56]; thus, the surface properties of the electrode under investigation can have a possible contribution in electrochemical results. Considering the values presented in Table 2, one can note that the roughness measured for TiSiCN-1 before corrosion examination proved a smother surface since the Ra parameter showed a value of ~473 nm, whereas for TiSiCN-2 R_a_, it was ~545 nm.

The high-frequency parameters Q_coat_ and *R*_pore_ represent the properties of the reactions at the electrolyte-coating interface. The results show that coating capacitance exhibits higher values for TiSiCN-coated Ti6Al4V, indicating a more active exchange with the NaCl solution, especially in the case of TiSiCN-2. Since there are insufficient data in the low-frequency region in the measured frequency range, it was not possible to determine the *R*_ct_ parameter. The *χ*^2^ parameter was also presented and the low values have indicated a validation of the fitting procedure.

The evolution of the open circuit potential (*E*_oc_) during the 1 h immersion is given in Figure 5a. Both coatings exhibited a constant evolution of *E*_oc_, indicating a good coating stability in 3.5% NaCl solution during 1 h immersion tests. Under stoichiometric coatings (TiSiCN-1) exhibited more electropositive *E*_oc_ values compared to the over stoichiometric coatings (TiSiCN-2). The uncoated alloy displayed an unstable surface due to the fluctuations, despite the fact that it exhibited electropositive value of *E*_oc_.

The potentiodynamic polarization curves of the investigated surfaces are shown in Figure 5b, while the electrochemical parameters are presented in Table 4. The resistance to corrosion of the investigated samples could be estimated according to the following principles:(1)Electropositivity signifies good resistance to corrosion: more electropositive corrosion potential value (*E*_corr_) means that the material is nobler in the used electrolyte. Figure 5b reveals that the most noble corrosion potential was measured with uncoated Ti6Al4V alloy. TiSiCN-2 coatings exhibit the more electronegative value, indicating poor corrosion resistance;(2)Surfaces with a low *i_corr_* value demonstrated a good corrosion resistance: all coatings presented lower *i_corr_* values compared to the uncoated Ti6Al4V alloy. TiSiCN-1 coating showed a lower *i_corr_* value compared to TiSiCN-2 coating, including the uncoated substrate;(3)Surfaces having higher anticorrosive properties demonstrate high polarization resistance (*R*_p_) values. Considering the *R*_p_ values in Table 4, note that both coatings exhibit higher *R*_p_ value than the uncoated substrate;(4)Porosity of the coatings was also considered: surfaces with low porosity have good anticorrosive properties. TiSiCN-1 coating has low porosity, indicating that the increase in C/N ratio leads to an increase in porosity, meaning the loss of corrosion resistance;(5)The protective efficiency (*P_e_*) was also considered: the highest *P_e_* value was obtained for the TiSiCN-1 coating.

To summarize the electrochemical investigations, the increase in C/N ratio leads to a loss of the anticorrosive properties of the TiSiCN coatings in 3.5% NaCl solution.

### 3.6. Neutron Activation Analysis for Determination of Fast Neutrons Flux Density

The neutron activation analysis (NAA) principle lies in the interplay between neutrons and the nuclei from the investigated material. It generates gamma radiation, and its intensity is being determined. In the present research, we measured the neutrons fluency through the samples, as well as the density of the neutron flux in a certain energetical diapason. The neutron spectrum was taken down via neutron activation satellite sample. Nickel (Ni) was activated to ^58^Co and its gamma activity was used for the estimation of the fast neutron flux density. What makes Ni appropriate is its initial capture cross-section from 1 MeV for neutrons. The reactions are:(4)N60i+fast neutron→triton+C58o
(5)N58i+fast neutron→protons+C58o

In Figure 6 shows the energy cross-sections relations of the reactions. Judging by the upper right graph, only reaction (5) takes place in the energy range that is of importance for us. The reaction cross-section on the plateau part of the diagram equals tenths of a barn. Reaction (4) starts at > 16 MeV with a cross-section of tenths of a millibarn, as illustrated in the lower right graph (Figure 6).

Despite the general rule that the energy should not surpass 3 MeV in laboratory conditions, registered X-ray and X-ray radiation energies vary between 40 keV to 10 MeV. The Canberra GC10021 spectrometer has integral nonlinearity of 0.025% and that distinctly points to an energy-channel adequacy; also, the FWHM is 1.1 keV for the 122 keV line and 1.8 keV for the 1332 keV line.

For the neutron flux densities estimation, an operative cross-section of σ_eff_ = 92 mb was employed, as well as considering the ^60^Ni nuclei to ^58^Ni nuclei ratio in the measurement wire and cross-sections integral convolutions of the fulfilled reactions. Figure 7 presents the NAA satellite gamma-spectrum after illumination with specimens.

The location of the total absorption peak at E_y_ = 810.7 keV matches ^58^Co (Figure 7). Its activity could be used in the fast neutrons flux density evaluation for each sample. Calculations determined the fast neutrons flux density and the fluency of both samples to be of 1.65 × 10^7^ n cm^−2^ s^−1^ and of 1.71 × 10^13^ n cm^−2^, respectively. The gamma dose rate was estimated via dosimeter after the specimen removal from the irradiation unit. Prompt gamma decay, rather than delayed beta decay, would indicate sample activation. The dosimeter background level indications of 0.1–0.2 μSv\h revealed no sample activation. The absence of nucleus with significant cross-section to interact with the fast neutrons in the studied sample predicts this aspect. Bearing in mind the vast fast neutron flux and the fluencies reached, the samples non-activation could indicate their potential use under reactor analogous conditions. That would be so since it is well known that the most severe defect formation process is caused by fast neutrons. Future studies regarding the resulting number of defects in the crystal structure of such are necessary.

## 4. Conclusions

TiSiCN carbonitride coatings prepared at different C/N ratios of 0.4 and 1.6, respectively, were developed by cathodic arc evaporation. The SEM investigation showed the presence of micro-droplets of different sizes present on the surfaces, especially for the high carbon coating. Some shallow oxides might form during the immersion, as indicated by XRD, since there was a decrease in the diffraction lines intensities after the corrosion assessment; however, there was no visible change in the SEM surface morphology. Modification of surface roughness as compared with the pre-tested coatings was observed mainly in the case of TiSiCN obtained at 1.6 C/N ratio (TiSiCN-2). According to EIS-fitted parameters, the highest pore resistance value was shown by TiSiCN-1 coating, indicating a more compact structure with beneficial effects on electrochemical behavior; also, a more uniform current distribution along the surface was observed in this case, which could be linked with a smoother surface. During samples immersion, more electropositive *E*_oc_ values, lower *i_corr_*, and a higher polarization resistance were exhibited by the under stoichiometric coatings (TiSiCN-1). On the other hand, the increase in C/N ratio leads to an increase in porosity and, consequently, to surfaces with lower anticorrosive properties. In the case of irradiation with 1.65 × 10^7^ n/cm^2^/sec fast neutron flux, short and long-lived isotopes were not observed in the samples, not even after 288 h. These results proved that both TiSiCN coatings can be successfully used in the severe conditions, which are characteristic to nuclear plants.

## Figures and Tables

**Figure 1 materials-16-01835-f001:**
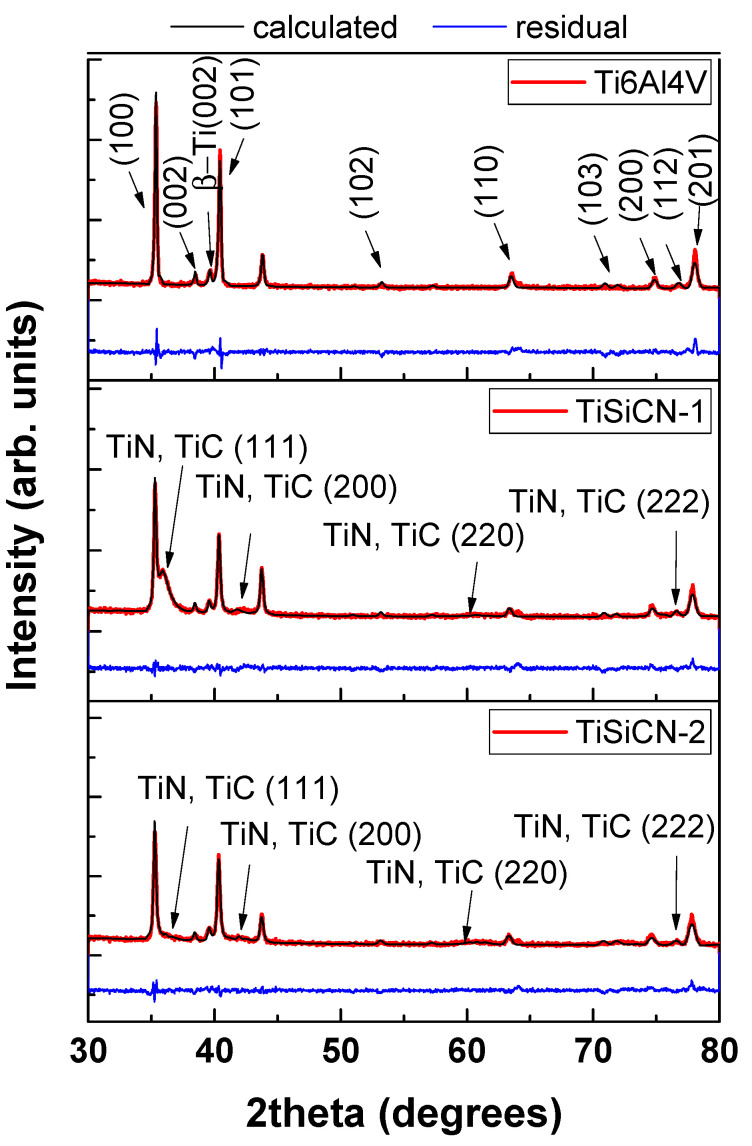
Rietveld fit of Ti6Al4V substrate (main phases—α-Ti [97.84 (2.16) wt.%] and β-Ti [2.16 (0.19) wt.%], TiSiCN-1, and TiSiCN-2.

**Figure 2 materials-16-01835-f002:**
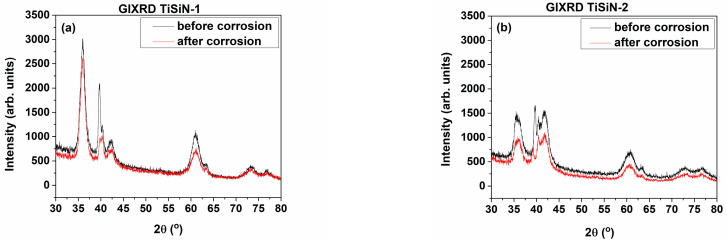
Comparison of GIXRD spectra before and after corrosion for: (**a**) TiSiCN-1 and (**b**) TiSiCN-2.

**Figure 3 materials-16-01835-f003:**
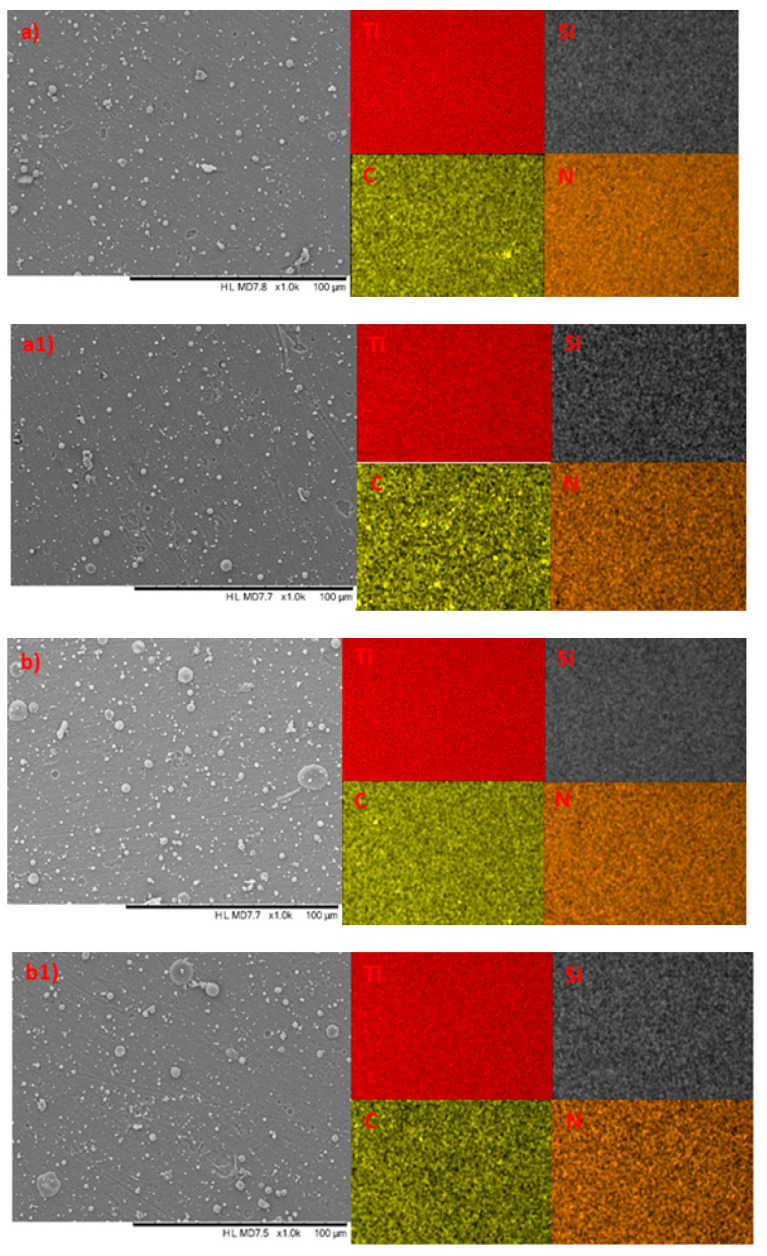
SEM images of TiSiCN coatings on Ti6Al4V alloy with elemental mapping profiles: (**a**) TiSiCN-1 before corrosion testing; (**a1**) TiSiCN-1 after corrosion; (**b**) TiSiCN-2 before corrosion testing; and (**b1**) TiSiCN-2 after corrosion.

**Figure 4 materials-16-01835-f004:**
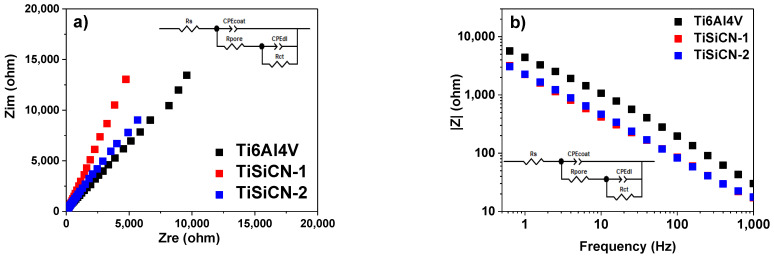
(**a**) Nyquist and (**b**) Bode amplitude plots for the investigated samples after 1 h immersion in 3.5% NaCl.

**Figure 5 materials-16-01835-f005:**
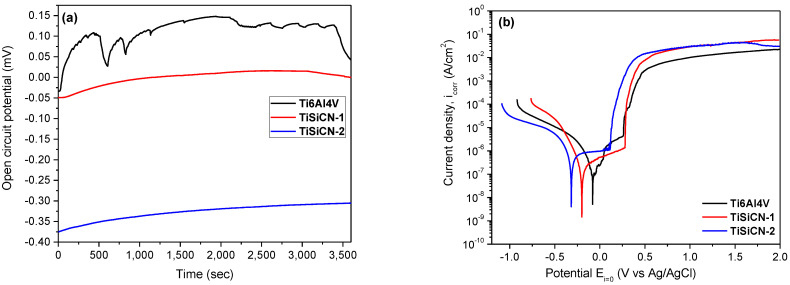
(**a**) Open circuit curves and (**b**) potentiodynamic polarization curves of the investigated surfaces after 1 h immersion in 3.5% NaCl.

**Figure 6 materials-16-01835-f006:**
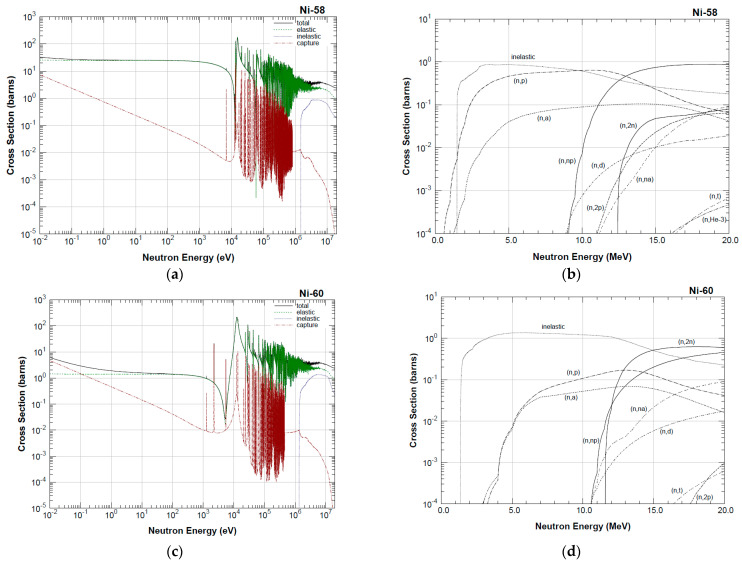
Energy dependences: (**a**,**c**)—effective nuclear cross-section of neutron interaction reactions with nuclei ^58^Ni and ^60^Ni; (**b**,**d**)—cross-sections of neutron capture reactions for ^58^Ni and ^60^Ni.

**Figure 7 materials-16-01835-f007:**
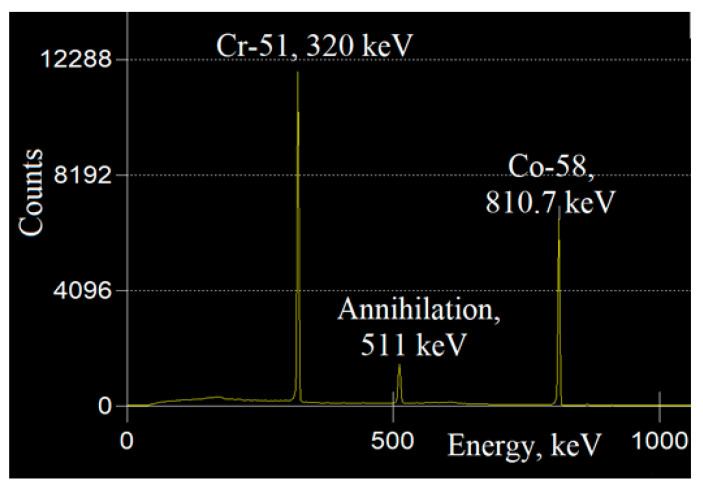
The measured gamma-spectrum of the NAA satellite after irradiation.

**Table 1 materials-16-01835-t001:** Elemental composition determined by EDS for TiSiCN-1 and TiSiCN-2 coatings.

	Elemental Composition (at.%)								
Coatings	Ti	Si	C	N	O	V	Al	(C + N)/Metal + Si	C/N
TiSiCN-1	42.30	5.17	15.91	34.36	2.12	0	0.14	0.61	0.46
TiSiCN-2	44.12	4.57	29.16	19.19	2.90	0	0.10	0.71	1.52

**Table 2 materials-16-01835-t002:** Roughness parameters of the investigated surfaces before and after corrosion tests (R_a_—arithmetic average deviation from the mean line; S_k_—skewness factor).

Substrate and Samples	Before Corrosion		After Corrosion	
	*R*_a_ (nm)	S_k_	*R*_a_ (nm)	S_k_
Ti6Al4V	36.69 ± 9.7	−0.11 ± 0.1	24.94 ± 2.7	−0.15 ± 0.1
TiSiCN-1	473.92 ± 19.0	1.99 ± 0.3	462.85 ± 35.3	1.71 ± 0.3
TiSiCN-2	545.55 ± 33.5	1.50 ± 0.3	544.59 ± 52.4	2.16 ± 0.6

**Table 3 materials-16-01835-t003:** The fitting results of EIS data for the investigated samples after 1 h immersion in 3.5% NaCl.

Sample	Rs (Ω cm^2^)	Qcoat (μF s^(α−1)^ cm^−2)^	α_coat_	*R*_pore_ (kΩ cm^2^)	*Q*_dl_ (μF s^(α−1)^ cm^−2^)	α_dl_	*R*_ct_ (kΩ cm^2^)	*χ^2^*
Ti6Al4V	9	76.17	0.53	4	5004	1.0	-	4.0 × 10^−4^
TiSiCN-1	8	89.65	0.77	3626	12.21	1.0	-	2.0 × 10^−4^
TiSiCN-2	5	120.71	0.63	17	10.65	0.98	-	4.0 × 10^−4^

**Table 4 materials-16-01835-t004:** Electrochemical parameters of the investigated samples after 1 h immersion in 3.5% NaCl: *E*_corr_: corrosion potential; *i_corr_*: corrosion current density; *R*_p_: polarization resistance; *P*: porosity; *P*_e_: protective efficiency.

Substrate and Samples	*E*_corr_ (mV)	*i_corr_* (nA)	*R*_p_ (Ω × 10^−3^)	*P_e_* (%)	*P*
Ti6Al4V	−78	222.123	0.195	-	-
TiSiCN-1	−199	25.281	8.795	88.6	0.016
TiSiCN-2	−316	94.347	1.816	57.5	0.058

## Data Availability

Not applicable.

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
