# Peer review of "TiSiCN as Coatings Resistant to Corrosion and Neutron Activation"

_materials, 2023, doi:10.3390/ma16051835_

Round 1
Reviewer 1 Report
The composition of the article meets the journal's recommendations and includes the following sections: Abstract, Introduction, Materials and Methods (divided into three sections), Results and discussion (divided into six sections), Conclusions and References. The aim of the work was to investigate the effect of neutron activation on TiSiCN carbonitride coatings prepared by cathodic arc evaporation method in a reactive gas mixture of C2H2 and N2. Moreover, mentioned coatings with a variable the C/N ratio, deposited on Ti6Al4V substrate, were tested in terms of elemental and phase composition, crystalline structure, morphology, corrosion resistance in aggressive saline solution (3.5% NaCl).
In the reviewer’s opinion, the article is very well prepared. The results of carried out tests confirm the hypotheses put forward by the Authors. It should also be noted that discussing the obtained results, Authors refer to a number of papers, within the scope of the work.
Finally, in my opinion, only a few weak points should be improved before the publication. The weaknesses are given below.
1. The quality of Figures 1,3 & 6 need to be improved.
2. Section 3.3 appears twice - incorrect numbering.
3. There are some typos in the article - examples below:
Lines 85:
Is: Si3N4
Should be: Si3N4
Lines 138:
Is: for CH4 and at 65 sccm for N2;
Should be: for CH4 and at 65 sccm for N2;
Author Response
Dear Ms. Marina-Bianca Bardas,
Thank you for your note and the reviewer comments on our manuscript. We would like to show our great gratitude to the editor and reviewer for the useful comments and constructive suggestions on our manuscript, which do help us significantly improve the quality of the current paper. All the review comments are appreciated. We have revised our manuscript accordingly. The revision of the paper was highlighted by the blue coloured font. Detailed and point-to-point response to the reviewer comments is summarized below.
Here, we re-submit a new version of our manuscript which has been checked and modified after our careful referring to the reviewer comments. Meanwhile, efforts were also made to improve the English of the paper. We hope all these changes will make this manuscript accepted by reviewers. Thank you for your kind consideration.
Best regards,
Alina Vladescu
- The quality of Figures 1,3 & 6 need to be improved.
The quality of the Figures 1, 3 and 6 was improved.
- Section 3.3 appears twice - incorrect numbering.
Thank you for pointing this out, the sections were renumbered.
- There are some typos in the article - examples below:
Lines 85:
Is: Si3N4
Should be: Si3N4
Lines 138:
Is: for CH4 and at 65 sccm for N2;
Should be: for CH4 and at 65 sccm for N2;
The authors appreciate the observation and modifications were carried out accordingly.

Reviewer 2 Report
1. Please adjust the frame of introduction to make it easier read for reader. Such as the mentioned lines 98-101 on page 2, it should be placed at the end of introduction as the concluding expression.
2. How long time do you immerse your sample in NaCl solution? As your mentioned “The comparison of XRD spectra before and after corrosion procedure in saline solution did not show any significant changes. However, for TiSiCN-2, we observe considerable decrease of intensities of the diffraction peaks” in lines 255-270 on page 6, is it sure? Are you sure that the decrease of intensities can be attributed to the occurrence of corrosive products (rutile)? There is no obvious evidence to verify this assumption, and the decrease of intensities can also be attributed to the error of detection.
3. For the surface morphology in Figure 3, the author wants to explain what happens before and after the corrosion test? I cannot find the specific conclusion except for some referred references. Please clarify.
4. As the mentioned lines 292-293 on page 7, the author quotes the structure of reference to analyze the corrosion change, but what is the real structure of these TiSiCN coatings? The XRD analysis in Figure 1 is also not specified.
5. As your mentioned “As observed in Nyquist impedance plots slightly 376 higher semicircles were obtained for TiSiCN-2 coating” in lines 376-377 of page 9, are you sure? Based on your data, the TiSiCN-1 should have the higher semicircles. In addition, I have a query why the substrate Ti6Al4V alloy has the highest Z value in Bode plot even if its semicircles is the lowest. In generally, do the high Z value represents for the excellent corrosion resistance?
6. I found that the frequency range of EIS is from 0.5 to 103 Hz. Do you ensure the accuracy of these data under the short measuring range? The author also referred “Since there are insufficient data in the low-frequency region in the measured frequency range, it was not possible to determine the Rct parameter” in lines 424-426 of page 11, so does the EIS detection need to be redone?
7. The language of this manuscript should be carefully improved, and some sentences are hard to understand.
Author Response
Dear Ms. Marina-Bianca Bardas,
Thank you for your note and the reviewer comments on our manuscript. We would like to show our great gratitude to the editor and reviewer for the useful comments and constructive suggestions on our manuscript, which do help us significantly improve the quality of the current paper. All the review comments are appreciated. We have revised our manuscript accordingly. The revision of the paper was highlighted by the blue coloured font. Detailed and point-to-point response to the reviewer comments is summarized below.
Here, we re-submit a new version of our manuscript which has been checked and modified after our careful referring to the reviewer comments. Meanwhile, efforts were also made to improve the English of the paper. We hope all these changes will make this manuscript accepted by reviewers. Thank you for your kind consideration.
Best regards,
Alina Vladescu
Reviewer 2 comments
- Please adjust the frame of introduction to make it easier read for reader. Such as the mentioned lines 98-101 on page 2, it should be placed at the end of introduction as the concluding expression.
Thank you for pointing this out. The introduction section was modified accordingly.
- How long time do you immerse your sample in NaCl solution? As your mentioned “The comparison of XRD spectra before and after corrosion procedure in saline solution did not show any significant changes. However, for TiSiCN-2, we observe considerable decrease of intensities of the diffraction peaks” in lines 255-270 on page 6, is it sure? Are you sure that the decrease of intensities can be attributed to the occurrence of corrosive products (rutile)? There is no obvious evidence to verify this assumption, and the decrease of intensities can also be attributed to the error of detection.
The samples were immersed 1h for the stabilization of the open circuit and ~5h for EIS and potentiodynamic curves to be recorded. We suspect that this reflection belongs to TiO2-rutile phase (200). Rutile, although not soluble in saline solution, apparently has worse adhesion with the rest of the coating layer and it is washed away during the corrosion treatment. Maybe the further investigation should be done, but we will have in visage for the next paper.
The decrease of intensities can not be attributed to the error of detection. We have repeated the measurements and we obtained the same spectrum. Thus, we assuming that the measurements were correctly performed.
Overall, higher decrease is observed in TiSiCN-2, implying higher corrosion resistance with increased N content. The decrease, however, is not uniform - strongest decrease is observed in intensity of peak at 2θ=39.6°. This is, however, subtle effect and we merely make assumption about rutile existence timidly without claiming we are completely sure. Must be repeated that this phase is product of deposition technique, not corrosion, the later eventually degrades it.
- For the surface morphology in Figure 3, the author wants to explain what happens before and after the corrosion test? I cannot find the specific conclusion except for some referred references. Please clarify.
As observed in SEM images, no visible evidence of a possible deterioration or the presence of corrosion products was observed on the surface of the samples after corrosion evaluation. This result indicates that the investigated coatings maintained their integrity after immersion, having a high corrosion resistance, as confirmed by electrochemical studies.
- As the mentioned lines 292-293 on page 7, the author quotes the structure of reference to analyze the corrosion change, but what is the real structure of these TiSiCN coatings? The XRD analysis in Figure 1 is also not specified.
Concerning microstructure characterization, to investigate properly micro-droplets on the surface, microdiffraction methods like SAED or CBED are needed, and it will be taking into account for the further experiments. Concerning crystal structure, it is described in the text. Structure of α-Ti (hexagonal) and β-Ti (cubic) is well known and taken from the literature. Concerning remark about non-specified Figs.1 - yes, probably here more detailed information has to be shown. For this purpose, some in line text is added to the figures. One can notice one of the substrate peaks cannot be described by Ti phases, program automatic search ascribes it to iron phase, and this could be some kind of smearing introduced during cutting process. This is not of great importance, since it is a pure substrate peak, not of the coating.
- As your mentioned “As observed in Nyquist impedance plots slightly 376 higher semicircles were obtained for TiSiCN-2 coating” in lines 376-377 of page 9, are you sure? Based on your data, the TiSiCN-1 should have the higher semicircles. In addition, I have a query why the substrate Ti6Al4V alloy has the highest Z value in Bode plot even if its semicircles is the lowest. In generally, do the high Z value represents for the excellent corrosion resistance?
There was a typing error, the reviewer is right, TSiCN-1 showed slightly higher semicircles and the typo was corrected in the manuscript. The interpretation of EIS data should be made by taking into consideration both Nyquist and Bode plots. Even though the substrate showed a high impedance modulus in the chosen frequency range, the fitting results of recorded impedance showed the lowest Rpore value in case of the thin oxide coating formed on top of Ti6Al4V alloy.
- I found that the frequency range of EIS is from 0.5 to 103Hz. Do you ensure the accuracy of these data under the short measuring range? The author also referred “Since there are insufficient data in the low-frequency region in the measured frequency range, it was not possible to determine the Rct parameter” in lines 424-426 of page 11, so does the EIS detection need to be redone?
It is well known the experimental noise contribution on the EIS data recorded at low frequencies. Considering the fact that a mandatory condition in order to obtain reliable results consist in having a good signal-to-noise ratio, the authors were forced to adjust the frequency range, since an erroneous fitting can lead to misleading assumptions, affecting inclusively the rest of the parameters which are characteristic to the coatings. Even though the lack of the data made Rct parameter impossible to be determined in the chosen frequency range, in this case, the statistical evaluation was the main weighting factor in order to ensure the reliability of the electrochemical parameters obtained following the EIS data fitting procedure.
- The language of this manuscript should be carefully improved, and some sentences are hard to understand.
Thank you for this suggestion. The authors revised the manuscript, and the language was improved.

Reviewer 3 Report
This paper prepared TiSiCN coatings on Ti6Al4V by using cathodic arc deposition to enhance the anti-corrosion property under irradiation conditions. The topic is meaningful, but it still needs to largely improved in terms of the following points.
1. Language should be carefully checked and revised. Too many mistakes!!
a) For instance, section 2.1, “The deposition conditions were selected to have: •Si content up to 6 at.%; •C/N ratio of 0.4 (under stoichiometric) and of 1.5 (over stoichiometric); •(C+N)/(metal+Si) ratio between 0.6 and 0.8. Depending on the C/N ratio, under stoichiometric coating (0.4) is labelled TiSiCN-1 and over stoichi-136 ometric (1.5) as TiSiCN-2.” This seems to be protocol-type statements.
b) Please check the upper and lower index in the whole manuscript.
c) Please define the abbreviation in the place where it appears for the first time. e.g. NAA.
d) In Line 172,” …performed over 0.5 ÷ 103 Hz frequency…”
e)In line 208, “For confidence”, is it right?
f) in line 260, “It this case” should be “In this case” ??
g) in line 502, “repectfully” should be “respectively” ??
……
2. Please check the equation (1) and (3). The form of equation (1) does not follow the standard demands for equation in published paper. In equation (3) and the following explanation, please check what is the difference between DEcorr and DEi=0.
3. The three spectra in Fig.1 were suggested to put in same figure in stacked way. The peaks are better to index in the figure, which will be more clear, it also can be applied in Figure 2.
4. Except the “droplets”, no more information can be observed from SEM result. So SEM at high magnification is strongly suggested to provided. And the thickness of the coating should be supplied.
5. The EIS measurements is doubtable!! Why is the range of frequency in 0.5-103Hz used? Normally the frequency of EIS is set as 105-0.01 Hz. The importance thing is that it can not be seen two time constants in Nyquist and Bode plots!! So why this EC in Fig.4 was used to fit the spectra? And no Rct was obtained from the fitting. So this fitting and explaination can not be acceptable for EIS results! Additionally, the relationship between αcoat and roughness in line 405-411 should be carefully stated!
6. Wrong statement. “more electropositive corrosion potential value (Ecorr) means that the material is nobler in the used electrolyte, indicating good corrosion resistance. “ Ecorr is one of thermodynamic parameters, it can not be suggested the corrosion rate or corrosion resistance, it only indicate the tendency of the reaction.
7. Figures 6 and 7 should be re-plotted with higher resolution. At current state, no values and data could be readable, especially for fig. 6.
Author Response
Dear Ms. Marina-Bianca Bardas,
Thank you for your note and the reviewer comments on our manuscript. We would like to show our great gratitude to the editor and reviewer for the useful comments and constructive suggestions on our manuscript, which do help us significantly improve the quality of the current paper. All the review comments are appreciated. We have revised our manuscript accordingly. The revision of the paper was highlighted by the blue coloured font. Detailed and point-to-point response to the reviewer comments is summarized below.
Here, we re-submit a new version of our manuscript which has been checked and modified after our careful referring to the reviewer comments. Meanwhile, efforts were also made to improve the English of the paper. We hope all these changes will make this manuscript accepted by reviewers. Thank you for your kind consideration.
Best regards,
Alina Vladescu
Reviewer 3 comments
Comments and Suggestions for Authors
This paper prepared TiSiCN coatings on Ti6Al4V by using cathodic arc deposition to enhance the anti-corrosion property under irradiation conditions. The topic is meaningful, but it still needs to largely improved in terms of the following points.
- Language should be carefully checked and revised. Too many mistakes!!
- a) For instance, section 2.1, “The deposition conditions were selected to have: •Si content up to 6 at.%; •C/N ratio of 0.4 (under stoichiometric) and of 1.5 (over stoichiometric); •(C+N)/(metal+Si) ratio between 0.6 and 0.8. Depending on the C/N ratio, under stoichiometric coating (0.4) is labelled TiSiCN-1 and over stoichi-136 ometric (1.5) as TiSiCN-2.” This seems to be protocol-type statements.
- b) Please check the upper and lower index in the whole manuscript.
- c) Please define the abbreviation in the place where it appears for the first time. e.g. NAA.
- d) In Line 172,” …performed over 0.5 ÷ 103 Hz frequency…”
e)In line 208, “For confidence”, is it right?
- f) in line 260, “It this case” should be “In this case” ??
- g) in line 502, “repectfully” should be “respectively” ??
The authors appreciate the observation and modifications were carried out accordingly.
- Please check the equation (1) and (3). The form of equation (1) does not follow the standard demands for equation in published paper. In equation (3) and the following explanation, please check what is the difference between DEcorr and DEi=0.
The authors verified the standard demands for equation in the “Instructions for authors” and equation (1) and (3) are in an editable format. DEi=0 was replaced with DEcorr in the figure (3) explanation.
- The three spectra in Fig.1 were suggested to put in same figure in stacked way. The peaks are better to index in the figure, which will be more clear, it also can be applied in Figure 2.
Thank you for this observation. We have corrected the figures 1 and 2.
- Except the “droplets”, no more information can be observed from SEM result. So SEM at high magnification is strongly suggested to provided. And the thickness of the coating should be supplied.
Thank you for this suggestion. As observed in SEM images, no visible evidence of a possible deterioration or the presence of corrosion products was observed on the surface of the samples after corrosion evaluation. This result indicates that the investigated coatings maintained their integrity after immersion and a higher magnification would not describe the overall corrosion surface.
As stated in the manuscript, the coating thickness was about 3 μm.
- The EIS measurements is doubtable!! Why is the range of frequency in 0.5-103Hz used? Normally the frequency of EIS is set as 105-0.01 Hz. And no Rct was obtained from the fitting. So this fitting and explaination can not be acceptable for EIS results!
It is well known the experimental noise contribution on the EIS data recorded at low frequencies. Considering the fact that a mandatory condition in order to obtain reliable results consist in having a good signal-to-noise ratio, the authors were forced to adjust the frequency range, since an erroneous fitting can lead to misleading assumptions, affecting inclusively the rest of the parameters which are characteristic to the coatings. Even though the lack of the data made Rct parameter impossible to be determined in the chosen frequency range, in this case, the statistical evaluation was the main weighting factor in order to ensure the reliability of the electrochemical parameters obtained following the EIS data fitting procedure.
The importance thing is that it can not be seen two time constants in Nyquist and Bode plots!! So why this EC in Fig.4 was used to fit the spectra?
The authors did attempt to simulate the EIS results of the substrate using a one-time constant, however, the error obtained indicated a poor agreement between the experimental and simulated data. According to literature, a bilayer structure formation was observed in the case of Ti-based alloys as a function of the testing solution [https://doi.org/10.1016/j.corsci.2021.109728], passive layer which is formed from Ti-based oxides and suboxides, as demonstrated by the XPS [https://doi.org/10.1016/S0142-9612(00)00145-9, https://doi.org/10.1016/j.corsci.2019.01.020, https://doi.org/10.1016/j.electacta.2007.12.041]. Considering these findings, a two-time constants EEC was selected in the current study for both the coated and the bare Ti alloy, the phase constants in this case being convoluted. Similar results were obtained by Gugelmin et al. [https://doi.org/10.1590/1516-1439.201514].
Additionally, the relationship between αcoat and roughness in line 405-411 should be carefully stated!
Thank you for this observation. Modifications were made related to the EIS fitted parameters, as follows:
“This coating also showed the highest αcoat parameter (0.77), followed by TiSiCN-2 (0.63), values related to non-uniform current distribution along the surface [55]. At the interface with the substrate, there is an almost defect free surface, since αdl showed high values, near 1. Thus, the constant phase element (CPE) used in this case could be seen as an ideal capacitor. Instead, at electrolyte- coating interface αcoat showed low values and a CPE is needed to take into consideration possible deviations from the ideal dielectric behavior [56]. As pointed out in literature, there are multiple factors associated with these deviations, which include surface disorder or inhomogeneity, geometric irregularities, working electrode porosity or ether surface roughness [56]. Thus, the surface properties of the electrode being under investigation can have a possible contribution in electrochemical results. Considering the values presented in Table 2, one can note that the roughness measured for TiSiCN-1 before corrosion examination proved a smother surface since Ra parameter showed a value of ~473 nm, whereas for TiSiCN-2 Ra, it was ~545 nm”.
- Wrong statement. “more electropositive corrosion potential value (Ecorr) means that the material is nobler in the used electrolyte, indicating good corrosion resistance. “ Ecorr is one of thermodynamic parameters, it can not be suggested the corrosion rate or corrosion resistance, it only indicate the tendency of the reaction.
According to many other researchers (Ecorr) means that the material is nobler. Thus, we deleted the part “indicating good corrosion resistance”.
- Figures 6 and 7 should be re-plotted with higher resolution. At current state, no values and data could be readable, especially for fig. 6.
Thank you for pointing this out. Figures 6 and 7 were re-plotted at higher resolution. We have also submitted as jpeg with high resolution.

Reviewer 4 Report
The authors of the current work reported on the fabrication, characterization, and evaluation of TiSiCN carbonitrides as protective coatings against corrosion in a 3.5% NaCl medium. The developed coatings were tested also for possible application in severe conditions such as a nuclear plant. Overall. this work is original, rich in data, and well-written and organized. I have only the following minor comments:
1. Why the authors did not consider increasing the thickness of the developed coating on the substrate? I believe 3 mm is relatively a small thickness for attaining excellent anti-corrosion properties.
2. I have a major concern about the electrochemical evaluation of the anti-corrosion performance of the samples:
-According to Figure 4, the substrate is showing higher impedance and larger semicircle compared to the fabricated coatings, which indicates a lower anticorrosion performance for the developed coatings compared to the substrate. Can the authors comment on this?
- Usually, different EEC is used to simulate the coating samples (two-time constant) compared to the bare substrate (one-time constant), which is not the case here. Have the authors attempted to simulate the EIS results of the substrate using a one-time constant ECC?
-I am not yet convinced why the authors failed to obtain and report the Rct values in Table 3. Can the authors elaborate more on this point?
-In Figure 5a, why the authors did not give more time to the OCP of the substrate sample to get stabilized?
-Kindly add the immersion time details to the captions and Tables of electrochemical data.
Author Response
Dear Ms. Marina-Bianca Bardas,
Thank you for your note and the reviewer comments on our manuscript. We would like to show our great gratitude to the editor and reviewer for the useful comments and constructive suggestions on our manuscript, which do help us significantly improve the quality of the current paper. All the review comments are appreciated. We have revised our manuscript accordingly. The revision of the paper was highlighted by the blue coloured font. Detailed and point-to-point response to the reviewer comments is summarized below.
Here, we re-submit a new version of our manuscript which has been checked and modified after our careful referring to the reviewer comments. Meanwhile, efforts were also made to improve the English of the paper. We hope all these changes will make this manuscript accepted by reviewers. Thank you for your kind consideration.
Best regards,
Alina Vladescu
Reviewer 4 comments
Comments and Suggestions for Authors
The authors of the current work reported on the fabrication, characterization, and evaluation of TiSiCN carbonitrides as protective coatings against corrosion in a 3.5% NaCl medium. The developed coatings were tested also for possible application in severe conditions such as a nuclear plant. Overall. this work is original, rich in data, and well-written and organized. I have only the following minor comments:
- Why the authors did not consider increasing the thickness of the developed coating on the substrate? I believe 3 mm is relatively a small thickness for attaining excellent anti-corrosion properties.
The main goal of the study was to investigate the influence of the C/N ratio on corrosion evaluation and neutron activation. Even the thickness was ~3 μm, the results indicated an excellent behaviour of the TiSiCN coatings under severe conditions, mostly attributed to the adatom mobility of the condensing species during deposition process characteristic to the selected evaporation method.
We believe that the thickness of 3 μm is a proper one. This value was selected based on our previous experience on thin films (more than 25 years) and based on some preliminary experiments. These preliminary experiments are not published, and we are not intended to publish, because they were used just to select the best deposition conditions for our propose. As an example, in the past, we had deposited a TiCN coating with thickness of 5 μm and we investigated the corrosion in 3.5%NaCl , and we found that the corrosion and wear was poor due to the low of adhesion between coating and substate.
- I have a major concern about the electrochemical evaluation of the anti-corrosion performance of the samples:
-According to Figure 4, the substrate is showing higher impedance and larger semicircle compared to the fabricated coatings, which indicates a lower anticorrosion performance for the developed coatings compared to the substrate. Can the authors comment on this?
The interpretation of EIS data should be made by taking into consideration both Nyquist and Bode plots and the fitting results. Even though the substrate showed a high impedance modulus in the chosen frequency range, the fitting results of recorded impedance showed the lowest Rpore value in case of the thin oxide coating formed on top of TiAlV alloy.
- Usually, different EEC is used to simulate the coating samples (two-time constant) compared to the bare substrate (one-time constant), which is not the case here. Have the authors attempted to simulate the EIS results of the substrate using a one-time constant ECC?
The authors did attempt to simulate the EIS results of the substrate using a one-time constant, however, the error obtained indicated a poor agreement between the experimental and simulated data. After a literature survey, a bilayer structure formation was observed in the case of Ti-based alloys as a function of the testing solution [https://doi.org/10.1016/j.corsci.2021.109728], passive layer which is formed from Ti-based oxides and suboxides, as demonstrated by the XPS [https://doi.org/10.1016/S0142-9612(00)00145-9, https://doi.org/10.1016/j.corsci.2019.01.020, https://doi.org/10.1016/j. electacta.2007.12.041]. Considering these findings, a two-time constants EEC was selected in the current study for both the coated and the bare Ti alloy. The statistical evaluation was the main weighting factor in order to ensure the reliability of the electrochemical parameters obtained following the EIS data fitting procedure.
-I am not yet convinced why the authors failed to obtain and report the Rct values in Table 3. Can the authors elaborate more on this point?
It is well known the experimental noise contribution on the EIS data recorded at low frequencies. Considering the fact that a mandatory condition in order to obtain reliable results consist in having a good signal-to-noise ratio, the authors were forced to adjust the frequency range, since an erroneous fitting can lead to misleading assumptions, affecting inclusively the rest of the parameters which are characteristic to the coatings. Even though the lack of the data made Rct parameter impossible to be determined in the chosen frequency range, in this case, the statistical evaluation was the main weighting factor in order to ensure the reliability of the electrochemical parameters obtained following the EIS data fitting procedure.
-In Figure 5a, why the authors did not give more time to the OCP of the substrate sample to get stabilized?
The time used for OCP was considered in order to accommodate all the investigated systems and it represents a minimum time for polarization technique to be performed. Both TiSiCN coatings showed a stable behaviour when immersed in 3.5 % NaCl, whereas in the same time frame Ti alloy showed potential fluctuation which can be ascribed to the nature of the passive layer formed.
-Kindly add the immersion time details to the captions and Tables of electrochemical data.
Thank you for this suggestion. The immersion time details and the used solution were added in the Figures and Tables caption.

Round 2
Reviewer 2 Report
The content of this paper has improved.